# Genetic Diversity and Phylogenetic Relationship of Human Norovirus Sequences Derived from Municipalities within the Sverdlovsk Region of Russia

**DOI:** 10.3390/v16071001

**Published:** 2024-06-21

**Authors:** Roman Bykov, Tarek Itani, Polina Starikova, Svetlana Skryabina, Anastasia Kilyachina, Stanislav Koltunov, Sergey Romanov, Aleksandr Semenov

**Affiliations:** 1Federal Budgetary Institution of Science, «Federal Scientific Research Institute of Viral Infections «Virome»», Federal Service for Surveillance on Consumer Rights Protection and Human Wellbeing, Ekaterinburg 620030, Russia; itani_tm@niivirom.ru (T.I.); starikova_pk@niivirom.ru (P.S.); semenov_av@niivirom.ru (A.S.); 2Federal Service for Surveillance on Consumer Rights Protection and Human Wellbeing in the Sverdlovsk Region, Ekaterinburg 620078, Russia; skryabina_sv@66.rospotrebnadzor.ru; 3Federal Budgetary Healthcare Institution «Center of Hygiene and Epidemiology in the Sverdlovsk Region», Ekaterinburg 620078, Russia; kilyachina_as@66.rospotrebnadzor.ru (A.K.); koltunov_sv@66.rospotrebnadzor.ru (S.K.); romanov_sv@66.rospotrebnadzor.ru (S.R.); 4Department of Medical Microbiology and Clinical Laboratory Diagnostics, Ural State Medical University, Ekaterinburg 620109, Russia; 5Institute of Natural Sciences and Mathematics, Ural Federal University Named after the First President of Russia B.N. Yeltsin, Ekaterinburg 620075, Russia

**Keywords:** human noroviruses, genotyping, phylogenetic analysis, genogroup GII, genetic distance

## Abstract

Human noroviruses (HuNoVs) are highly contagious pathogens responsible of norovirus-associated acute gastroenteritis (AGE). GII.4 is the prevailing HuNoV genotype worldwide. Currently there are no studies on the molecular monitoring and phylogenetic analysis of HuNoVs in the territory of the Sverdlovsk region; therefore, it is not possible to objectively assess their genetic diversity. The aim of the study is to carry out genotyping and phylogenetic analysis of HuNoVs in the Sverdlovsk region from 2022 to 2023. Fecal samples (*n* = 510) were collected from children suffering from HuNoV-AGE in municipalities of the Sverdlovsk region and the capsid genotype was determined by amplifying the ORF1/ORF2 junction. Of the 196 HuNoVs typed, which represent 38% of the studied samples, the largest share of HuNoV genotypes belong to the GII genogroup—86%, followed by the GI genogroup—14%. Noroviruses GII.4 and GII.17 were the co-dominant capsid genotypes (33.2% each). Phylogenetic analysis demonstrates that the identified sequences on the territory of the Sverdlovsk region have the smallest genetic distance, which gives grounds for their unification into a common cluster. Routine monitoring and phylogenetic analysis of circulating norovirus pathogens spectrum will enable timely tracking of HuNoVs genetic diversity and evolutionary events. This will lead to the development of more effective anti-epidemic measures, ultimately reducing the burden of infectious diseases.

## 1. Introduction

Human Noroviruses (HuNoVs) are responsible for approximately one fifth of all cases of acute non-bacterial gastroenteritis, causing an estimated 699 million illnesses and over 200,000 deaths [1,2]. HuNoVs can lead to both sporadic cases of infection in organized communities and large-scale epidemics [3]. HuNoV infections are ubiquitous, affecting all age groups, with higher rates of illness and mortality observed in children and older adults [4,5]. Two genogroups, GI and GII, are most frequently associated with human norovirus gastroenteritis. The GII genetic group alone accounts for over 50% of global outbreak cases. The predominant HuNoV genotype worldwide is GII.4 [6,7,8,9]. Due to the high frequency of recombination variability, new variants of GII.4 replace each other every 2 to 3 years, rapidly spreading among the global population, creating favorable conditions for the spread and increase in morbidity rates within the human population [10].

Noroviruses belong to the genus *Norovirus*, family *Caliciviridae*, and are a genetically diverse group of viruses that infect a wide range of mammalian hosts. The genome of noroviruses consists of a single-stranded (+) RNA, which contains three main open reading frames (ORFs) that control the synthesis of viral proteins: ORF1, which encodes the RNA-dependent RNA polymerase (RdRp) and other replication cycle proteins; ORF2, which encodes the major capsid protein VP1; and ORF3, which encodes the minor capsid protein VP2 [11]. One of the key functions is performed by the structural protein VP1, which consists of two domains, the S domain and the P domain [12]. The S domain is the capsid shell that participates in the formation of icosahedral symmetry of the capsid. The P domain, on the other hand, is divided into two subdomains, P1 and P2, which carry antigenic determinants and determine the specificity of binding to cell surface polysaccharide complexes [13,14]. The susceptibility of an individual to HuNoVs is determined by the expression of genetically determined antigens of the human blood group (HBGA) [15,16]. These glycoconjugates serve as initial co-receptors that are necessary for the adsorption and penetration of the norovirus virion into the host cell [17,18]. According to the scientific literature, it is the functionally active *FUT2* gene that controls the expression of the enzyme alpha-1,2-fucosyltransferase 2, which is responsible for the formation of the HBGA complex and, as a result, the development of genetically determined resistance to HuNoVs gastroenteritis. The presence of one functional allele in the *FUT2* gene leads to the potential synthesis of the active enzyme alpha-1,2-fucosyltransferase 2, resulting in a high susceptibility to norovirus [19].

According to official reports «On the state of sanitary and epidemiological well-being of the population in the Russian Federation in 2022», the incidence rate of human norovirus infection in Russia was 29.28 per 100,000 population, which is 2.3 times higher than the average multi-year period from 2010 to 2019 (13.25). The Sverdlovsk region is characterized by high terms of incidence rates of HuNoV AGE (119.75) and ranks third among the subjects of the Russian Federation. The volume of implementation of molecular biological methods in laboratory diagnostics had a significant impact on the incidence of HuNoV infections. During the period 2020–2022, anti-epidemic measures aimed at preventing COVID-19 also played a key role according to a State Report by Rospotrebnadzor, the Federal Service for Surveillance on Consumer Rights Protection and Human Wellbeing, titled «Sanitary and Epidemiological Well-being in the Russian Federation in 2021» [20]. The utilization of a standardized genotyping system for HuNoVs and subsequent phylogenetic analysis will allow for a comprehensive assessment of the genetic diversity of HuNoVs and track events of evolutionary divergence within the viral population to control the incidence of infection in endemic territories of the Sverdlovsk region. In this study, we aimed to perform genotyping and phylogenetic analysis of a HuNoV strains circulating identified strains in individual municipalities of the Sverdlovsk region from 2022 to 2023.

## 2. Materials and Methods

### 2.1. Stool Samples

From February 2022 to December 2023, fecal samples were collected from children (range: 2–7 years) with HuNoV AGE in municipalities of the Sverdlovsk region with high disease incidence, such as Ekaterinburg, Kamensk-Uralsky, Sukhoi Log, Verkhnyaya Pyshma, Pervouralsk, Revda, and Nizhny Tagil (Figure 1).

All received biological samples were screened for the presence of HuNoVs by using an enzyme immunoassay (Norovirus-antigen-enzyme immunoassays-Best, Vector-Best, Novosibirsk, Russian Federation) or qPCR (AmpliSens^®^ Rotavirus/Norovirus/Astrovirus-FL, Central Research Institute of Epidemiology, Moscow, Russian Federation).

### 2.2. Ethics Statements

This research has obtained approval from the Local Ethics Committee at the Federal Budgetary Institution of Science «Federal Scientific Research Institute of Viral Infections «Virome» Federal Service for Surveillance on Consumer Rights Protection and Human Wellbeing Rospotrebnadzor» (Protocol № 1 dated 17 March 2022). Fecal samples were collected from medical institutions of the Sverdlovsk region: The Ekaterinburg Consultative Diagnostic Center and the State Autonomous Healthcare Institution of the Sverdlovsk region Children’s City Hospital (Kamensk-Uralsky). All patients gave their written consent to participate in the study, and patient data were stored anonymously and securely. All fecal samples were transported to the Federal Scientific Research Institute of Viral Infections «Virome» and stored at −20 °C.

### 2.3. Isolation of RNA, PCR, and Sequencing

A 10% suspension in physiological saline was prepared from native fecal samples, vortexed, and clarified by centrifugation for 3 min at 10,000× *g*, from which nucleic acids were extracted using the Riboprep^®^ kit (Central Research Institute of Epidemiology, Moscow, Russian Federation). Subsequently, reverse transcription was performed to obtain cDNA from RNA templates using the REVERTA-L kit (Central Research Institute of Epidemiology, Moscow, Russian Federation) according to the manufacturer’s protocol. To amplify the norovirus pathogen, specific genome regions corresponding to the ORF1/ORF2 region were selected using the degenerate primers G1SKF/G1SKR and G2SKF/G2SKR [21]. The reaction mixture was prepared using 5X ScreenMix (Evrogen, Moscow, Russian Federation) and primers for GI (G1SKF, G1SKR) and GII genogroups (G2SKF, G2SKR) [21]. The amplification protocol consisted of an initial step at 94 °C for 5 min, followed by 45 cycles with denaturation at 94 °C for 30 s, annealing at 50 °C for 30 s, and extension at 72 °C for 2 min, and a final extension step at 72 °C for 7 min. DNA extraction from the gel was carried out using the PureLink™ Quick Gel Extraction and PCR Purification Combo Kit (Invitrogen, Lifetechnologies, Carlsbad, CA, USA). The PCR products were subsequently sequenced in both directions using the same primers for PCR and the BigDye™ Terminator v3.1 Cycle Sequencing Kit (Applied Biosystems, Austin, TX, USA) on an ABI 3130 Genetic Analyzer (Applied Biosystems, Austin, TX, USA).

### 2.4. Phylogenetic Analysis

The obtained genetic sequence was identified using the BLAST service (ABI 3130 DNA Analyzer). To create consensus sequences, norovirus reference sequences from NCBI exhibiting the highest homology with the typed sample were selected. To generate consensus nucleotide sequences based on the results of sequences obtained from both forward and reverse reads, we used the UGENE software, version 47 for DNA and protein sequence visualization, alignment, assembly, and annotation [22]. The number of nucleotides from the consensus sequences included in the phylogenetic analysis for GI was 300 bp while for GII it was 290 bp In the phylogenetic analysis, 12 consensus sequences of GI and 24 sequences of GII were included. The genetic distances matrix and the multiple alignment algorithm (ClustalW) were utilized in the MEGA software, version 11, to examine the genetic sequences of HuNoVs [23]. The neighbor-joining method, together with the Kimura-2 parameter model, was used to create phylogenetic trees and determine pairwise distances between taxa in two norovirus genogroups. The reliability of the additive phylograms’ topology was assessed using 1000 bootstrap replications. Statistical support values greater than 70 are indicated on the phylograms. The assessment of evolutionary divergence between sequences was conducted using the genetic analysis MEGA software, version 11. Distance estimation was performed on amino acid sequences aligned with the ClustalW algorithm. The model for determining genetic distances involves examining the proportion of nucleotide or amino acid sites (p/a) where two compared sequences are different. This comparison is achieved by dividing the number of p/a differences by the total number of compared p/a, expressing the divergence as a percentage. The nucleotide sequences obtained in this study were submitted to GenBank. The total number of sequences currently deposited is 81, with accession numbers OP862431.1 to OP862440.1; ON681575.1 to ON681586.1; OP862363.1 to OP862368.1; OP862428.1 to OP862430.1; OR447701.1 to OR447707.1; OR685664.1 to OR685666.1; OR794008.1; OR717542.1 to OR717547.1; OR726224.1; OR751620.1 to OR751621.1; OR399135.1 to OR399146.1; OR399514.1; OR426510.1 to OR426514.1; OR447699.1 to OR447700.1; and OR399123.1 to OR399134.1

## 3. Results

### 3.1. Distribution of HuNoV Genotypes in the Sverdlovsk Region from 2022 to 2023

A total of *n* = 510 samples of clinical material were analyzed during the study period (*n* = 440 positive by enzyme immune assay and *n* = 70 positive by qPCR). Of these, 196 HuNoVs were successfully typed by Sanger sequencing, accounting for 38% (196/510) of the total number of analyzed samples. It was determined that over the analyzed period, the GII genogroup was predominant—*n* = 168, 86%, while genogroup GI accounted for *n* = 28, 14%. Molecular genetic monitoring identified a variety of norovirus genotypes in the territories of the individual municipalities. The distribution of the identified genotypes significantly differed between of 2022 and 2023. In 2022, a significant proportion of the detected norovirus genotypes were the GII.17 (*n* = 32/76, 42%), GI.3 (*n* = 17/76, 22%), and GII.4 (*n* = 7/76, 9%) genotypes. In 2023, the genotype profile was mainly represented by GII.4 (*n* = 58/120, 48%), GII.17 (*n* = 33/120, 28%), and GI.2 (*n* = 2/120, 1.6%). In 2023, the genotype GI.3 was not identified in the Sverdlovsk region (Figure 2).

### 3.2. Phylogenetic Analysis of HuNoV Strains Belonging to Genogroup GI

The phylogenetic analysis was conducted on rare HuNoV genotypes, GI.2, GI.3, GI.5, and GI.6, which have not been previously reported in the Sverdlovsk region (Figure 3). On the additive phylogram, all identified strains in the Sverdlovsk region form common clusters within each genotype, indicating the lowest genetic distance between each other. Each cluster, constructed based on strains identified in the municipalities of Ekaterinburg and Kamensk-Uralsky, shows both mono- and polyphyletic connections with strains from Brazil, Japan, Thailand, South Africa, the USA, Spain, China, and India, indicating widespread distribution of the identified HuNoV genotypes.

For a more accurate assessment of genetic distance, a nucleotide base distance matrix was constructed in the GI.3 cluster. The terminal clade with GI.3 ancestors diverges into two branches with bootstrap values above 90. One of the branches forms the GI.3 cluster with sequences from Ekaterinburg, showing the lowest genetic distance within the cluster. On the second branch, a sequence GI.3/3090 from Kamensk-Uralsky is observed, exhibiting the highest genetic distance from the formed GI.3 cluster from Ekaterinburg. During amino acid sequence alignment, seven missense mutations were detected in genotype GI.3/3039 compared to the reference sequences of GI.3 from Ekaterinburg. The analysis of the genetic distance matrix showed a significant 8.1% divergence in the amino acid se-quences in GI.3/3039, with several mutations appearing (Figure 4).

### 3.3. Phylogenetic Analysis of HuNoV Strains Belonging to Genotypes GII.4 and GII.17

Phylogenetic analysis of HuNoV strains belonging to the most frequent genotypes GII.4 and GII.17 was conducted (Figure 5). The phylogenetic analysis demonstrates that the GII.4 viral strains identified in the Sverdlovsk region exhibit low genetic distance and form a single cluster. The GII.4 cluster displays polyphyletic relationships with viral strains from China, the USA, and Thailand. The GII.17 strains detected in the Sverdlovsk region form a cluster, with close phylogenetic connections to sequences from China, Japan, and Thailand. The viral sequences of genotypes GII.17 and GII.4 display pronounced genetic homogeneity within their respective clusters. In the GII.4 and GII.17 clusters, viral strains from the municipalities of Kamensk-Uralsky, Ekaterinburg, and Sukhoi Log exhibit low divergence rates—2.1% for the GII.4 cluster and 2.8% for the GII.17 cluster.

## 4. Discussion

Our study provides information on the genetic diversity and molecular–epidemiological characteristics of HuNoVs circulating in six separate municipalities of the Sverdlovsk region over a two-year period from 2022 to 2023. The pandemic period of COVID-19 has contributed significantly to a decrease in the registration of other infectious pathogens, including human norovirus infection [24]. This may be attributed to the mobilization of laboratory resources for SARS-CoV-2 testing [25,26,27] as well as the implementation of personal hygiene measures on the backdrop of COVID-19 restrictive measures [28,29]. Since 2022, there has been a resurgence in norovirus infection rates in Russia and other countries, bringing it back to prepandemic levels [30]. The rise in norovirus infection cases was also affected by reallocating major lab resources for monitoring norovirus gastroenteritis in 2022, amid a decrease in overall COVID-19 illness rates. This postpandemic period demonstrates an identical genotypic profile in Russia and other countries, where the second genetic group, GII, is the dominant one—86%, while GI accounts for 14% [31,32,33,34]. The spectrum of circulating HuNoV genotypes has changed during the two-year surveillance of HuNoVs in the Sverdlovsk region. In 2022, GI was more frequently isolated in patients when compared to 2023. Previously dominant HuNoV genotypes, such as GII.4 and GII.17, were accompanied by the identification of rare GI.3, GI.5, and GI.6 genotypes, which were not previously reported in the Sverdlovsk region, nor deposited sequences in GenBank, NCBI. In 2022, we noted a 22% detection rate for genotype GI.3 while this genotype was not detected in 2023. By 2023, GII completely outcompeted GI, and the GII.4 genotype now occupies a leading position in the genotypic structure, as in many other countries [31,35], with the exception of norovirus genotype GI.2, detected in two patients in the vicinity of Ekaterinburg. The GII.17 genotype is widely circulating in the Sverdlovsk region. All Russian GII.17 sequences form polyphyletic connections with strains from China and Japan [36,37,38], where a similar new strain, GII.17[P17], was the main cause of numerous HuNoV outbreaks [39].

Eastern European countries, including Russia, are characterized by a unique genotype distribution of norovirus AGE agents over different time periods. In Bulgaria, the dominating strains belong to the GII.4, GII.20, and GII.3 genotypes, and no norovirus genotypes belonging to the GI genogroup were reported [40]. In addition, HuNoV monitoring in Hungary revealed that the most widely circulating strains belong to the GII.4 Lordsdale and GIIb Hilversum strains. A smaller proportion of detected genotypes included GII.1, GII.2, GII.7, GI.2, GI.3, GI.4, and GI.6 [41]. Moreover, the study on HuNoVs’ genetic diversity in Albania found that the GII.4 genotype is the prevailing genotype, with other reported viral strains belonging to the GII.2 and GII.3 genotypes [42]. This widespread dominance of GII.4 and its variants in various countries confirms that GII.4 is the most predominant genotype among noroviruses worldwide. Furthermore, studying the genetic diversity of other HuNoV genotypes in specific Eastern European countries reveals the unique distribution of noroviruses within different genogroups/genotypes in each studied area.

The dominant norovirus genotypes in specific districts of Germany have become variants of GII.4 (GII.4 [P16], GII.4 [P31]), GII.6 (GII.6 [P7]), and GI.4 (GI.4 [P4]), which correlates with the results obtained in the Sverdlovsk region in 2023. It is important to note that the strains identified in Germany were identified using the method of sequencing two reading frames, highlighting [P] types, while the results obtained in the Sverdlovsk region do not include [P] types [43]. Research on norovirus genetic diversity in Belgium shows a similar distribution of genotypes compared to the data obtained in the Sverdlovsk region for the years 2022–2023, where the dominant genotypes are GII (GII.4, GII.3, GII.2) and GI (GI.3, GI.4, GI.5, GI.6) [44].

A study by Anfruns-Estrada on norovirus genetic diversity in Spain identified six predominant genotypes for GI [45]. In another study on the circulation of HuNoVs in Spain from 2016 to 2020, researchers detected a significant number of GI.3 strains. Between 2018 and 2020, they identified several GI.3 genovariants, including GI.3 [P3] at 35%, GI.3 [P13] at 17%, and GI.3 [P10] at 6%. These findings correlate with the detection rate of GI.3 (22%) in the Sverdlovsk region, sparking substantial scientific interest and calling for further open discussion [46]. When constructing the additive phylogram, the GI.3 [P10] genotype sequences exhibit the smallest genetic distance in relation to a potential novel genovariant GI.3, which has been identified in the Sverdlovsk region. The amino acid sequence genetic distance matrix of GI.3/3039 and GI.3 [P10]/3718 shows a 0% divergence, indicating their complete identity. A plausible scenario could involve the importation of the Spanish genovariant GI.3 [P10]/3718 onto the territory of Russia in 2022. Greece’s genotypic profile features dominant noroviruses from genotypes GII.4, GII.2, GII.6, GII.3, and two strains of GI.1 [47]. Therefore, current data on the distribution of norovirus strains in Western and Southern Europe indicate that the norovirus landscape largely resembles the genotypic profile identified in our study.

The proportion of successfully typed samples confirmed by ELISA was 34% (*n* = 153/440), while that of the successfully typed samples by qPCR was 61% (*n* = 43/70). The relatively low detection rate of norovirus agents in fecal samples can be attributed to the fact that the majority of positive collected samples were screened by an enzyme immunoassay method. The low specificity of the enzyme immunoassay method can lead to false positive results when detecting HuNoVs. The low percentage of genotyped HuNoVs may be due to a high percentage of false positive results in biological samples detected by the enzyme immunoassay method. In order to explain the low typed percentage, we screened 45 untyped specimens from 2023 by real-time PCR (AmpliSens^®^ Rotavirus/Norovirus/Astrovirus-FL, Central Research Institute of Epidemiology, Moscow, Russian Federation). Screening of these specimens revealed that only 13 (28%) samples were positive for norovirus infection agents, with extremely high cycle threshold values (Ct > 37.5), showing low viral load. The remaining 32 (72%) specimens were identified as rotaviruses, and thus excluded from this study.

According to published studies, amino acid sequences (aa) that differ from previous genotypes by 8% or more are considered new genotypes [48]. The potentially new endemic variant GI.3/3090 forms paraphyletic links with strains from Thailand in 2017–2019, which are its closest relatives [32,49]. The percentage of difference between GI.3/3090 and other GI.3 sequences from Ekaterinburg is 8.1%, which may indicate a possible divergent evolution and the circulation of a new endemic variant in the Sverdlovsk region [48]. All strains of the variant to which GI.3/3039 belongs have substitutions of S/A, E/D, A/V, A/V, N/S, Y/F, and H/Q, which may indicate the possible importation of this genotype to the Sverdlovsk region. To support this hypothesis on the emergence of a new endemic variant GI.3/3090, it can be said that attempts to trace the importation of norovirus infection agents are ineffective and unpromising, as there is low referral to medical care facilities and specialists in cases of HuNoV infections (7.5–10.1%) [50], as well as low manifestation of clinical symptoms.

This study has several limitations. First, we only detected HuNoV infections in hospitalized children with symptomatic AGE but not in asymptomatic children. Second, no samples were collected from adults (average age: 6 years, range: 2–7 years).

Therefore, the trends reported here might not be generalizable to other age groups. Third, we did not compare variables and detect differences between genotype profiles with respect to the rare genotypes, which are of extreme interest because they may represent potential emerging or zoonotic genotypes with consequences for public health. Last, international comparison of norovirus strains is complicated because of their genetic diversity and the use of different protocols and HuNoV genome fragments in genotyping; consequential differences result in sequences with diverging lengths and from diverging genomic regions.

## 5. Conclusions

For the first time on the territory of certain municipalities in the Sverdlovsk region, genetic analysis of HuNoV AGE was conducted. The HuNoV genotyping system based on amplification of the ORF1/ORF2 region allows successful identification of various HuNoV genotypes. The detected distribution of HuNoVs by genotypes and genogroups confirms that the highest proportion worldwide is attributed to the second genetic group GII. The genotype profile for the analyzed period of 2022–2023 is mainly represented by HuNoV genotypes GII.4 and GII.17. Rare norovirus genotypes, GI.3, GI.5, GI.6, and GI.2, were also identified and submitted to Genbank, NCBI. Genetic diversity monitoring of HuNoVs across Eastern European countries reveals significant divergence from the genotype profile observed in the Sverdlovsk region, highlighting the unique genetic distribution in Russia. Phylogenetic analysis demonstrates the lowest genetic distance between sequences of HuNoVs strains circulating in Russia and the formation of polyphyletic connections between sequences from China, the United States, Japan, India, and others. Further application of the presented genotyping framework will allow for greater accessibility in determining the genetic diversity of HuNoV infection agents in the Sverdlovsk region. Sequencing of HuNoV genomes and phylogenetic analysis will enable to investigate the potential infection source and identification of possible transmission factors in outbreaks. Routine monitoring is crucial for identifying circulating HuNoV genotypes that may subsequently be included in vaccine formulations. The data obtained from this study and other similar studies will allow to monitor and to establish a collection of strains that can be used in future studies to develop new treatments and vaccines.

## Figures and Tables

**Figure 1 viruses-16-01001-f001:**
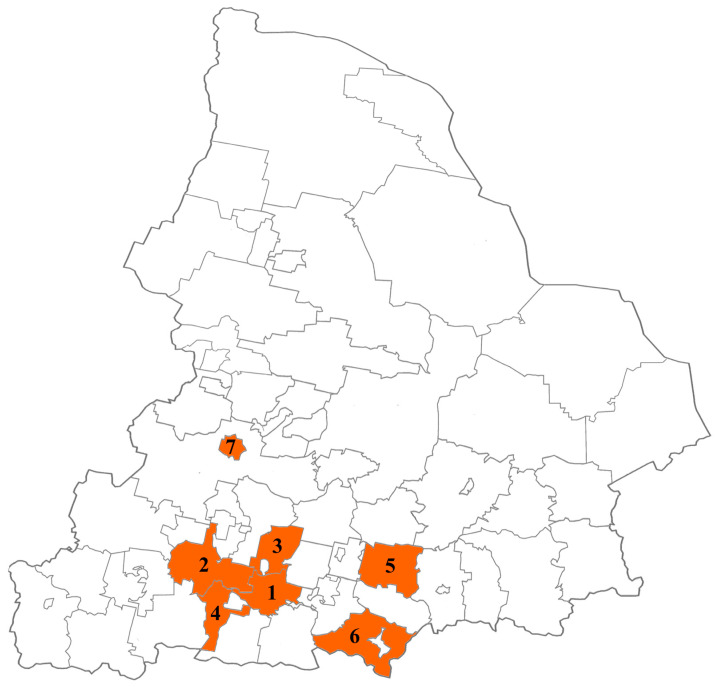
Municipalities of the Sverdlovsk region participating in this study (1—Ekaterinburg, 2—Pervouralsk, 3—Verkhnyaya Pyshma, 4—Revda, 5—Sukhoi Log, 6—Kamensk-Uralsky, 7—Nizhny Tagil).

**Figure 2 viruses-16-01001-f002:**
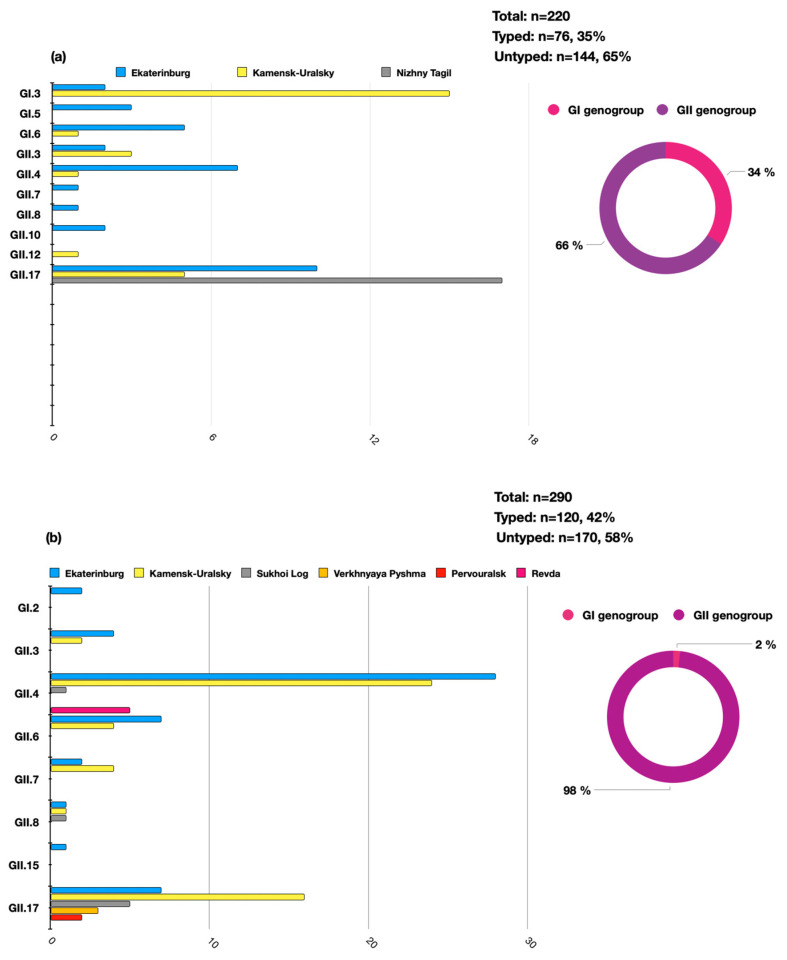
The distribution of the HuNoV genotypes in the Sverdlovsk region in 2022–2023 per genogroup (circular charts) and per genotype (histogram charts) ((**a**)—the distribution of HuNoVs in 2022, (**b**)—the distribution of HuNoVs in 2023).

**Figure 3 viruses-16-01001-f003:**
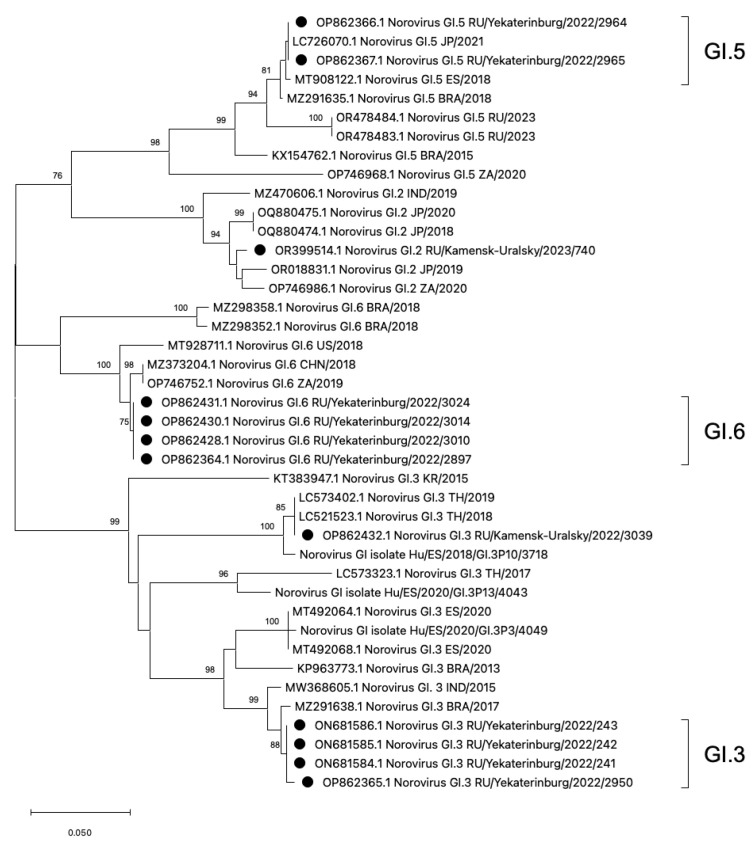
Phylogenetic tree based on the HuNoVs’ viral protein 1 major capsid gene fragment, constructed based on nucleotide sequences of HuNoV genotypes GI.5, GI.6, GI.2, and GI.3 (represented by black dots). Reference strains were downloaded from GenBank and labelled with their accession numbers, followed by genotypes, country, and year. The neighbor-joining phylogenetic tree was constructed with MEGA X software v. 11 and bootstrap tests (1000 replicates), based on the Kimura two-parameter model. Bootstrap values above 70% are given at branch nodes.

**Figure 4 viruses-16-01001-f004:**
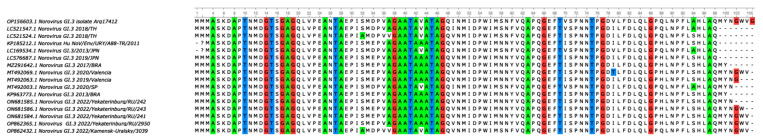
To analyze the amino acid sequences of genotype GI.3/3039, a multiple alignment was performed. The figure depicts a multiple alignment performed using the ClustalW algorithm and the ClustalW alignment matrix. The strain GI.3/3039 from the municipality of Kamensk-Uralsky exhibits several amino acid substitutions, which result in missense mutations, with amino acid substitutions at positions S > A; E > D; A > V; A > V; N > S; Y > F; and H > Q.

**Figure 5 viruses-16-01001-f005:**
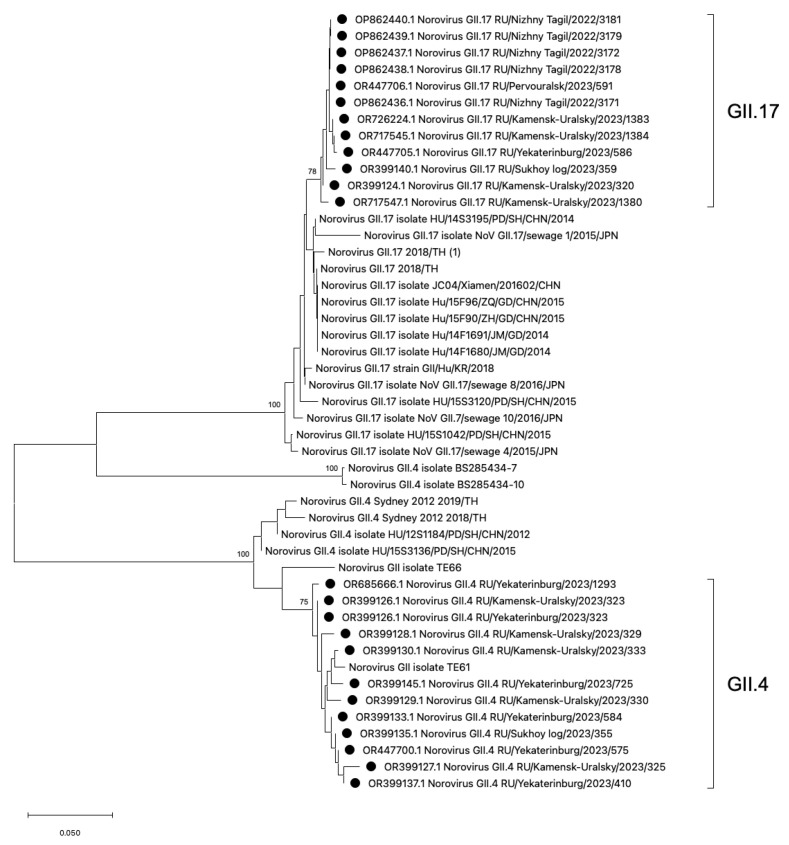
Phylogenetic tree based on the HuNoVs’ viral protein 1 major capsid gene fragment, constructed based on nucleotide sequences of the HuNoV genotypes GII.4 and GII.17 (represented by black dots). Reference strains were downloaded from GenBank and labelled with their accession numbers, followed by genotypes, country, and year. The neighbor-joining phylogenetic tree was constructed with MEGA X software v. 11 and bootstrap tests (1000 replicates), based on the Kimura two-parameter model. Bootstrap values above 70% are given at branch nodes.

## Data Availability

Dataset available on request from the authors.

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
