# Peer review of "Genetic Diversity and Phylogenetic Relationship of Human Norovirus Sequences Derived from Municipalities within the Sverdlovsk Region of Russia"

_viruses, 2024, doi:10.3390/v16071001_

Round 1

Reviewer 1 Report (New Reviewer)

Comments and Suggestions for Authors

The Bykov et al. manuscript describes the genotypes and phylogenetic analysis of human norovirus isolated from acute gastroenteritis episodes in a hospitalized young population. The main findings of this study reveal that noroviruses GII.4 and GII.17 were the co-dominant capsid genotype and most of the sequences identified showed a small genetic distance. The study is well-designed and has relevance to monitor the circulation of norovirus strains and track evolutionary events. There are minor revisions that need to be addressed before acceptance for publication:

Reference 19 is not well suited for the statement cited.

Pg 6: The legend for Figure 3 should be placed in Figure 4 and vice versa.

Figure 5 should be moved and described in the Results section.

Author Response

Dr. Eric O. Freed                                                         Ekaterinburg, June 17, 2024

Editor-in-Chief

Viruses MDPI journal 

Dear Dr. Freed,

We carefully studied each comment provided by the reviewers and responded accordingly in the text below. We appreciate the attention given to us and look forward to hearing back from you soon.

Comments from the Editors and Reviewers:

Reviewer 1: The Bykov et al. manuscript describes the genotypes and phylogenetic analysis of human norovirus isolated from acute gastroenteritis episodes in a hospitalized young population. The main findings of this study reveal that noroviruses GII.4 and GII.17 were the co-dominant capsid genotype and most of the sequences identified showed a small genetic distance. The study is well-designed and has relevance to monitor the circulation of norovirus strains and track evolutionary events. There are minor revisions that need to be addressed before acceptance for publication:

  1. Reference 19 is not well suited for the statement cited.

Response:

Thank you for your review and valuable suggestions. We have added a new publication corresponding to the data on FUT which is more suited to the statement.  Nazaret Peña-Gil; Santiso-Bellón, C.; Gozalbo-Rovira, R.; Buesa, J.; Monedero, V.; Jesús Rodrı́guez-Dı́az The Role of Host Glyco-biology and Gut Microbiota in Rotavirus and Norovirus Infection, an Update. International Journal of Molecular Sciences 2021, 22, 13473–13473. Lines [68-70]

  1. Pg 6: The legend for Figure 3 should be placed in Figure 4 and vice versa.

Response:

Thanks for your feedback regarding our results presentation clarity. Figures three and four take their respective places. Please check: Lines [192-193] for Figure 3, Lines [228-229] for Figure 5. Please note that figure 4 is now figure 5.

  1. Figure 5 should be moved and described in the Results section.

Response:

Thank you for pointing out the need for clarity in that section. Figure 5 has been relocated to the Results section, and is now numbered as figure 4 Lines [210-211].

On behalf of all the authors, thank you for your kind consideration!

Roman O. Bykov, Postgraduate student of the Federal Budgetary Institution of Science State Scientific Center of Virology and Biotechnology "Vector"

Junior Research Assistant in the Laboratory of Enteric Viral Infections.

Ekaterinburg Research Institute of Viral Infections,

Federal Budgetary Institution of Science «Federal Scientific Research Institute of Viral Infections «Virome» Federal Service for Surveillance on Consumer Rights Protection and Human Wellbeing Rospotrebnadzor,

620030, Ekaterinburg, Russian Federation +7(982) 690-69-64

Reviewer 2 Report (New Reviewer)

Comments and Suggestions for Authors

Conducting epidemiological surveillance of viruses causing acute gastroenteritis, such as norovirus, is of utmost importance, as they are responsible for a significant number of deaths worldwide, particularly among children under 5 years old, immunocompromised individuals, and the elderly. Therefore, this article is relevant as it provides information about the strains circulating in regions where norovirus surveillance was not previously conducted.

Below are some suggestions and requests for clarifications and minor revisions regarding the wording.

Line 18: To maintain consistent formatting throughout the text, it is recommended to use the plural form of the acronym HuNoVs, as it is written on line 38.

Lines 63 to 67: Please review this information. The presence of at least one functional FUT2 allele promotes the expression of the FUT2 enzyme and, consequently, the synthesis of the H antigen, resulting in the secretor phenotype, which is associated with increased susceptibility to HuNoV infection. You can include the following reference, which clearly explains this biochemical pathway: doi:10.3390/ijms222413473

Lines 118 to 120: It is not clear whether the amplification targeted the ORF1/ORF2 junction or only a portion of ORF2. Please specify which primers were used if the intention was to amplify the ORF1/ORF2 junction region. The primers G1SKF/G1SKR and G2SKF/G2SKR are designed to amplify a portion of the ORF2 region (VP1) and not ORF1. Therefore, as described in this study, only a partial amplification of the VP1 gene located in the ORF2 region has been performed.

Lines 127 to 129: The primers used for sequencing the PCR products are not specified. Was sequencing performed only in the forward direction? In the reverse direction? Or in both directions? If sequencing was performed in both directions, please indicate in the following paragraph (lines 131-133) which program was used to obtain the consensus sequence for each sample.

Lines 131 to 133: The consensus sequence is obtained by assembling the sequences obtained from both forward and reverse reads. Reference sequences are employed for alignment and subsequent phylogenetic analysis. Please review this description, and if the assembly of both reads was performed, please specify the program used for it.

Lines 146 to 148: Please specify the exact number of sequences published in GenBank, as there are some errors detected in the accession numbers. For example, the ranges between the accession numbers OR717542.1-OR751621.1 or OR399123.1-OR794008.1 are incorrect, as they correspond to 34,080 and 394,886 samples, respectively. Additionally, I have confirmed that within the range OP862363.1-OP862440.1, there are sequences that do not belong to norovirus. An example would be the sample OP862369.1. Please review all accession numbers accordingly.

Lines 160 to 161: It would be interesting to indicate what percentage of GI.3 samples were isolated in 2023, as in 2022 it represents a significant proportion.

Figure 2: Please review this figure or clarify in the text. In this figure, the number of genotyped samples is shown as 220 (Figure 2a) and 290 (Figure 2b). According to the description, genotyping was only performed on a total of 196 samples, distributed as 76 (year 2022) and 120 (year 2023). If you prefer to show the total number of samples analyzed, you should include in the graph the proportion of untyped samples.

Line 168: Twelve sequences are observed included in the tree. Please specify how many sequences were included in the phylogenetic analysis and why that selection was made from the total sequenced samples. This can be addressed in the Materials and Methods section. Similarly, provide this information for the analysis of the GII genogroup.

Line 169: The tree in Figure 3 corresponds to the analysis of the GII genogroup. The phylogenetic analysis of the GI genogroup is presented in Figure 4. Please make the necessary modifications.

Line 192: Please indicate the number of nucleotides included in the phylogenetic analysis of each of the genogroups. This information can be provided in the captions of Figures 3 and 4 or in the Materials and Methods section.

Lines 217 to 218: The phrase "in Russia... since 2020" could be omitted due to redundancy with the previous sentence.

Lines 221 to 222: In line 216, you indicate that the COVID-19 pandemic era led to a decrease in the reporting of other infectious pathogens. As described in lines 221-222, it appears that you attribute the increase in clinical cases to natural norovirus infection. However, it is plausible that, following the decrease in COVID-19 cases, resources were reallocated to monitor norovirus infection in 2022, resulting in an increase in case reporting.

Line 228: In 2022, the GI.3 genotype was isolated in 22% of cases, so it would be advisable to mention in the discussion the notable decrease in the proportion of GI.3 samples in the year 2023.

Line 235: Please use the new nomenclature established since 2019. The previous version, for example, GI.P6-GI.6, has been updated to GI.6[P6]. doi: 10.1099/jgv.0.001318

Line 261: Another study conducted in Spain covers a broader range of analyzed years. Additionally, this study mentions that the GI.3 genotype was one of the most predominant within the GI genogroup in the year 2020 (during the COVID-19 pandemic), representing 20% of the analyzed samples (a finding similar to your results in 2022). This discussion should be taken into consideration since in their study, the GI.3 genotype represented 22% of the analyzed samples in 2022. doi: 10.1128/spectrum.02505-21

Lines 265 to 266: There is a grammatical error. Please rewrite them.

Lines 282-284: In the previously mentioned publication (doi: 10.1128/spectrum.02505-21), two new genotypes within the GI genogroup are proposed after analyzing the genetic distance between several strains classified as GI.3. In this case, they relied on the Criteria of 2 times the standard deviation (2XSD) proposed by Chhabra P, et al. 2019 (doi.org/10.1099/jgv.0.001318). It would be interesting to include some of these strains in your phylogenetic analysis to determine if your strains could correspond to a new variant of GI.3 or rather to a new genotype within the GI genogroup. If they cluster with any of these strains, it should be cited in the text.

Line 313: As previously mentioned, please clarify the primers used for amplification of the ORF1/ORF2 junction region, if applicable, in the Materials and Methods section.

Author Response

Dr. Eric O. Freed                                                         Ekaterinburg, June 17, 2024

Editor-in-Chief 

Viruses MDPI journal 

Dear Dr. Freed,

We carefully studied each comment provided by the reviewers and responded accordingly in the text below. We appreciate the attention given to us and look forward to hearing back from you soon.

Comments from the Editors and Reviewers:

Reviewer 2: Conducting epidemiological surveillance of viruses causing acute gastroenteritis, such as norovirus, is of utmost importance, as they are responsible for a significant number of deaths worldwide, particularly among children under 5 years old, immunocompromised individuals, and the elderly. Therefore, this article is relevant as it provides information about the strains circulating in regions where norovirus surveillance was not previously conducted.

  1. Line 18: To maintain consistent formatting throughout the text, it is recommended to use the plural form of the acronym HuNoVs, as it is written on line 38.

Response:

Thank you for your review and valuable suggestions. We have changed the singular form to plural throughout the entire text of the article (HuNoVs).

  1. Lines 63 to 67: Please review this information. The presence of at least one functional FUT2 allele promotes the expression of the FUT2 enzyme and, consequently, the synthesis of the H antigen, resulting in the secretor phenotype, which is associated with increased susceptibility to HuNoV infection. You can include the following reference, which clearly explains this biochemical pathway: doi:10.3390/ijms222413473

Response:

Thank you for your meticulous review and valuable comments on our study. The text now reads:

“The presence of one functional allele in the FUT2 gene leads to the potential synthesis of the active enzyme alpha-1,2-fucosyltransferase 2, resulting in a high susceptibility to norovirus”. Lines [68-70]

The reference (Nazaret Peña-Gil et al, 2021) suggested by the reviewer has been added to the manuscript (reference 19).

  1. Lines 118 to 120: It is not clear whether the amplification targeted the ORF1/ORF2 junction or only a portion of ORF2. Please specify which primers were used if the intention was to amplify the ORF1/ORF2 junction region. The primers G1SKF/G1SKR and G2SKF/G2SKR are designed to amplify a portion of the ORF2 region (VP1) and not ORF1. Therefore, as described in this study, only a partial amplification of the VP1 gene located in the ORF2 region has been performed.

Response:

We thank the reviewer for this point. To monitor norovirus circulation routinely, we selected the ORF1/ORF2 region for PCR amplification. For the bioinformatic analysis, the amplified section of ORF1 that encodes RdRp was excluded due to its length and its lack of informativeness in phylogenetic analysis. The text now reads:

“To amplify the norovirus pathogen, specific genome regions corresponding to the ORF1/ORF2 region were selected using degenerate primers G1SKF/G1SKR and G2SKF/G2SKR”. Lines [121-123]

  1. Lines 127 to 129: The primers used for sequencing the PCR products are not specified. Was sequencing performed only in the forward direction? In the reverse direction? Or in both directions? If sequencing was performed in both directions, please indicate in the following paragraph (lines 131-133) which program was used to obtain the consensus sequence for each sample.

Response:

Thank you for your valuable suggestion. For sequencing the PCR product, we utilized primers flanking the ORF1/ORF2 region, G1SKF/G1SKR and G2SKF/G2SKR. The sequencing stage was performed in both directions. The text now reads:

The PCR products were subsequently sequenced in both directions using the same primers for PCR and the BigDye™ Terminator v3.1 Cycle Sequencing Kit (Applied Biosystems, Austin, USA) on an ABI 3130 Genetic Analyzer (Applied Biosystems, USA)”. Lines [130-133]

  1. Lines 131 to 133: The consensus sequence is obtained by assembling the sequences obtained from both forward and reverse reads. Reference sequences are employed for alignment and subsequent phylogenetic analysis. Please review this description, and if the assembly of both reads was performed, please specify the program used for it.

Response:

Thank you very much for your comment. The text now reads:

“To generate consensus nucleotide sequences based on the results of sequences obtained from both forward and reverse reads, we used the UGENE software for DNA and protein sequence visualization, alignment, assembly, and annotation”. Lines [137-140]

  1. Lines 146 to 148: Please specify the exact number of sequences published in GenBank, as there are some errors detected in the accession numbers. For example, the ranges between the accession numbers OR717542.1-OR751621.1 or OR399123.1-OR794008.1 are incorrect, as they correspond to 34,080 and 394,886 samples, respectively. Additionally, I have confirmed that within the range OP862363.1-OP862440.1, there are sequences that do not belong to norovirus. An example would be the sample OP862369.1. Please review all accession numbers accordingly.

Response:

Thank you for pointing out the need for clarity in that section. The text now reads:

“In the GenBank database, the total number of sequences currently deposited is 81 with accession numbers OP862431.1 to OP862440.1; ON681575.1 to ON681586.1; OP862363.1 to OP862368.1; OP862428.1 to OP862430.1; OR447701.1 to OR447707.1; OR685664.1 to OR685666.1; OR794008.1; OR717542.1 to OR717547.1; OR726224.1; OR751620.1 to OR751621.1; OR399135.1 to OR399146.1; OR399514.1; OR426510.1 to OR426514.1; OR447699.1 to OR447700.1; OR399123.1 to OR399134.1 Lines [156-162]

  1. Lines 160 to 161: It would be interesting to indicate what percentage of GI.3 samples were isolated in 2023, as in 2022 it represents a significant proportion.

Figure 2: Please review this figure or clarify in the text. In this figure, the number of genotyped samples is shown as 220 (Figure 2a) and 290 (Figure 2b). According to the description, genotyping was only performed on a total of 196 samples, distributed as 76 (year 2022) and 120 (year 2023). If you prefer to show the total number of samples analyzed, you should include in the graph the proportion of untyped samples.

Response:

Thank you for your careful review and suggestion. In 2023, the second genogroup GII almost completely displaced genogroup GI. Due to this reason, the genotype GI.3 was not detected in the Sverdlovsk region in 2023. The text now reads:

“In 2023, the genotype GI.3 was not identified in the Sverdlovsk region”. Lines [176-177]

Thank you for the comments regarding the histogram; all corrections have been made and are reflected in the figure. Lines [178-179]

  1. Line 168: Twelve sequences are observed included in the tree. Please specify how many sequences were included in the phylogenetic analysis and why that selection was made from the total sequenced samples. This can be addressed in the Materials and Methods section. Similarly, provide this information for the analysis of the GII genogroup.

Line 169: The tree in Figure 3 corresponds to the analysis of the GII genogroup. The phylogenetic analysis of the GI genogroup is presented in Figure 4. Please make the necessary modifications.

Response:

Thanks for your feedback regarding our results presentation clarity. Consensus sequences of two genogroups were added taking into account high-quality sequences post-assembly onto the reference sequence. It was also decided to exclude sequences with high similarity among them from the dataset, as this may lead to a weak phylogenetic signal, resulting in reduced support indices for the tree nodes. The necessary modification were made for Figures 3 and 4, which take their respective places. The text now reads:

In the phylogenetic analysis, 12 consensus sequences of GI and 24 sequences of GII were included. Lines [142-143].

  1. Line 192: Please indicate the number of nucleotides included in the phylogenetic analysis of each of the genogroups. This information can be provided in the captions of Figures 3 and 4 or in the Materials and Methods section.

Response:

Thank you for the valuable suggestions for complementing our research. The text now reads: “The number of nucleotides from the consensus sequences included in the phylogenetic analysis for GI was 300 b.p., while for GII it was 290 b.p.”

This text has been added to the material and methods section. Lines [140-142]

  1. Lines 217 to 218: The phrase "in Russia... since 2020" could be omitted due to redundancy with the previous sentence.

Response:

Thank you for your insightful comments. We completely agree with you, this sentence structure has been modified. The text now reads:

“The pandemic period of COVID-19 has contributed significantly to a decrease in the registration of other infectious pathogens, including human norovirus infection. This may be attributed to the mobilization of laboratory resources for SARS-CoV-2 testing as well as the implementation of personal hygiene measures on the backdrop of COVID-19 restrictive measures”. Lines [242-244]

  1. Lines 221 to 222: In line 216, you indicate that the COVID-19 pandemic era led to a decrease in the reporting of other infectious pathogens. As described in lines 221-222, it appears that you attribute the increase in clinical cases to natural norovirus infection. However, it is plausible that, following the decrease in COVID-19 cases, resources were reallocated to monitor norovirus infection in 2022, resulting in an increase in case reporting.

Response:

Thank you for your valuable comments. The text now reads:

“The rise in norovirus infection cases was also affected by reallocating major lab resources for monitoring norovirus gastroenteritis in 2022, amid a decrease in overall COVID-19 illness rates”. Lines [246-248]

  1. Line 228: In 2022, the GI.3 genotype was isolated in 22% of cases, so it would be advisable to mention in the discussion the notable decrease in the proportion of GI.3 samples in the year 2023.

Response:

Thank you for your thorough review and valuable comments on our manuscript. We agree on incorporating this wording into the discussion. The text now reads:

“In 2022, we noted a 22% detection rate for genotype GI.3,while this genotype was not detected in 2023”. Lines [256-257]

  1. Line 235: Please use the new nomenclature established since 2019. The previous version, for example, GI.P 6GI.6, has been updated to GI.6[P6]. doi: 10.1099/jgv.0.001318

Response:

Thank you for pointing out the issue with the nomenclature. The genotypes have been revised in accordance with the new nomenclature.  Lines [262; 279]

  1. Line 261: Another study conducted in Spain covers a broader range of analyzed years. Additionally, this study mentions that the GI.3 genotype was one of the most predominant within the GI genogroup in the year 2020 (during the COVID-19 pandemic), representing 20% of the analyzed samples (a finding similar to your results in 2022). This discussion should be taken into consideration since in their study, the GI.3 genotype represented 22% of the analyzed samples in 2022. doi: 10.1128/spectrum.02505-21

Lines 265 to 266: There is a grammatical error. Please rewrite them.

Response:

Thank you for your detailed comments. This research sparks scientific interest, and its results' interpretation will be part of the discussion in our article. The text now reads:

“In another study on the circulation of HuNoVs in Spain from 2016 to 2020, researchers detected a significant number of GI.3 strains. Between 2018 and 2020, they identified several GI.3 genovariants, including GI.3 [P3] at 35%, GI.3 [P13] at 17%, and GI.3 [P10] at 6%. These findings correlate with the detection rate of GI.3 (22%) in the Sverdlovsk region, sparking substantial scientific interest and calling for further open discussion”. Lines [288-293]

  1. Lines 265 to 266: There is a grammatical error. Please rewrite them.

Response: Thank you very much for pointing out the grammatical issues in our manuscript. The text now reads:

“The proportion of successfully typed samples confirmed by ELISA was 34% (n=153/440), while the successfully typed samples by qPCR was 61% (n=43/70)”. Lines [303-304]

  1. 16. Lines 282-284: In the previously mentioned publication (doi: 10.1128/spectrum.02505-21), two new genotypes within the GI genogroup are proposed after analyzing the genetic distance between several strains classified as GI.3. In this case, they relied on the Criteria of 2 times the standard deviation (2XSD) proposed by Chhabra P, et al. 2019 (doi.org/10.1099/jgv.0.001318). It would be interesting to include some of these strains in your phylogenetic analysis to determine if your strains could correspond to a new variant of GI.3 or rather to a new genotype within the GI genogroup. If they cluster with any of these strains, it should be cited in the text.

Response:

We express our sincere gratitude for the valuable suggestions aimed at enhancing the results of our research. Our GI.3 strains cluster with the strains from Spanish study by Noemi Navarro-Lleó.The new results obtained will be prominently featured in the discussion section. The text now reads:

“When constructing the additive phylogram, the GI.3 [P10] genotype sequences exhibit the smallest genetic distance in relation to a potential novel genovariant GI.3, which has been identified in the Sverdlovsk region. The amino acid sequence genetic distance matrix of GI.3/3039 and GI.3 [P10]/3718 shows a 0% divergence, indicating their complete identity. A plausible scenario could involve the importation of the Spanish genovariant GI.3 [P10]/3718 onto the territory of Russia in 2022”. Lines [293-298]

  1. 17. Line 313: As previously mentioned, please clarify the primers used for amplification of the ORF1/ORF2 junction region, if applicable, in the Materials and Methods section.

Response:

Thank you for your feedback, the remarks have been considered and incorporated into the Materials and Methods section.

On behalf of all the authors, thank you for your kind consideration!

Roman O. Bykov, Postgraduate student of the Federal Budgetary Institution of Science State Scientific Center of Virology and Biotechnology "Vector"

Junior Research Assistant in the Laboratory of Enteric Viral Infections.

Ekaterinburg Research Institute of Viral Infections,

Federal Budgetary Institution of Science «Federal Scientific Research Institute of Viral Infections «Virome» Federal Service for Surveillance on Consumer Rights Protection and Human Wellbeing Rospotrebnadzor,

620030, Ekaterinburg, Russian Federation +7(982) 690-69-64

This manuscript is a resubmission of an earlier submission. The following is a list of the peer review reports and author responses from that submission.

Round 1

Reviewer 1 Report

Comments and Suggestions for Authors

Dear authors. This is an interesting study aimed at the analysis of genetic diversity for human norovirus infection agents.

Although the paper is well-written and includes representative data that can be of interest for readers, I have some comments and suggestions that can improve its quality.

1. I suggest comparing the results obtained by the authors including the most prevalent isolates with those known for other regions (for instance, Europe).

2. What are the results of search for noroviruses sequences in freely-available databases of nucleotide and protein databases. Can the authors estimate the similarity between sequences of viruses available in the databases and those obtained in their own study.

3. Were the sequences of norovirus genome collected by the authors uploaded to any freely-available  repository? I think it would be extremely useful for the analysis of their structure by other scientists.

Comments on the Quality of English Language

English should be checked with the help of native speakers.

Author Response

Comments from the Editors and Reviewers:

Reviewer 1: Dear authors. This is an interesting study aimed at the analysis of genetic diversity for human norovirus infection agents. Although the paper is well-written and includes representative data that can be of interest for readers, I have some comments and suggestions that can improve its quality.

  1. I suggest comparing the results obtained by the authors including the most prevalent isolates with those known for other regions (for instance, Europe).

Response:

We thank the reviewer for this point. We have included a selection of recent studies from Europe, and compared the results of these studies with our results. This has been added to the text and highlighted in yellow in the article. The text now reads (Lines 251-262):

“The dominant norovirus genotypes in specific districts of Germany (Märkisch-Oderland) have become dominant variants GII.4 (GII.P16-GII.4, GII.P31-GII.4), GII.6 (GII.P7-GII.6), and GI.4 (GI.P4-GI.4), which correlates with the results obtained in the Sverdlovsk region in 2023 (Niendorf et al, 2023). It is important to note that the strains identified in Germany were identified using a dual typing method including VP1 and RdRP genes, while the results obtained in our study only included VP1 typing.  Research on norovirus genetic diversity in Belgium shows a similar genotype distribution compared to the data obtained in the Sverdlovsk region in 2022 and 2023, where the dominant genotypes were GII (GII.4, GII.3, GII.2), GI (GI.3, GI.4, GI.5, GI.6). (Elke Wollants et al, 2015). The study of norovirus genetic diversity in Spain identified six predominant genotypes for GI (GI.1, GI.2, GI.3, GI.4, GI.5, and GI.6) and five for GII (GII.2, GII.4, GII.6, GII.10, GII.17). (Anfruns-Estrada et al, 2022). Greece's genotypic profile features dominant noroviruses from genotypes GII.4, GII.2, GII.6, GII.3, and two strains of GI.1. Therefore, current data on the distribution of norovirus strains in Western and Southern Europe indicate that the norovirus landscape largely resembles the genotypic profile identified in our study. (Siafakas et al, 2023.)

Niendorf, S.; Faber, M.; Tröger, A.; Hackler, J.; Jacobsen, S. Diversity of Noroviruses throughout Outbreaks in Germany 2018. Viruses 2020, 12, 1157.

Wollants, E.; De Coster, S.; Van Ranst, M.; Maes, P. A Decade of Norovirus Genetic Diversity in Belgium. Infection, Genetics and Evolution 2015, 30, 37–44.

Anfruns-Estrada, E.; Sabaté, S.; Razquin, E.; Cornejo Sánchez, T.; Bartolomé, R.; Torner, N.; Izquierdo, C.; Soldevila, N.; Coronas, L.; Domínguez, À.; et al. Epidemiological and Genetic Characterization of Norovirus Outbreaks That Occurred in Catalonia, Spain, 2017-2019. Viruses 2022, 14, 488.

Nikolaos Siafakas; Anastassopoulou, C.; Lafazani, M.; Genovefa Chronopoulou; Rizos, E.; Pournaras, S.; Athanasios Tsakris Pre-dominance of Recombinant Norovirus Strains in Greece, 2016–2018. Microorganisms 2023, 11, 2885–2885.

  1. What are the results of search for noroviruses sequences in freely-available databases of nucleotide and protein databases. Can the authors estimate the similarity between sequences of viruses available in the databases and those obtained in their own study.

Response:

We utilized GenBank (NCBI database) for bioinformatic analysis and constructing additive phylograms of norovirus sequences. (Lines 132-133) To create consensus sequences, norovirus reference sequences from NCBI exhibiting the highest homology with the typed sample were selected. The search results featured closely related norovirus strains from different countries exhibiting high identity levels (Per. ident 98.00-100.00%) and significant coverage (Query Cover 95.00-100.00%).

  1. Were the sequences of norovirus genome collected by the authors uploaded to any freely-available repository? I think it would be extremely useful for the analysis of their structure by other scientists.

Response:

We thank the reviewer for this comment. The nucleotide sequences obtained in this study were submitted to GenBank, NCBI accession numbers : OR717542.1 – OR751621.1; ON681575.1 – ON681586.1; OR399123.1 – OR794008.1; OP862363.1 – OP862440.1. These numbers already featured in the text (Lines 146-148).

Reviewer 2 Report

Comments and Suggestions for Authors

The authors analyzed norovirus sequences derived from stool samples of infants with AGE in specific municipalities of Russian Sverdlovsk region. Molecular NoV epidemiology data of that regions have not been published yet.

“Genetic Diversity and Phylogenetic Analysis for Human 2 Norovirus Infection Agents in Specific Municipalities within The Sverdlovsk Region of Russia”  is not correct

I would rather write:

“Genetic Diversity and Phylogenetic relationship of human norovirus sequences derived from Municipalities within The Sverdlovsk Region of Russia”

In line 90,91 you state: “Positive biological samples were confirmed through laboratory testing using enzyme immunoassays”

With immunoassays you detect only samples with higher viral loads (ca. > 10e5 Geq/ml). This means that your samples are selected

What percentage of the immunoassay positive samples have you successfully sequenced?

Please describe Fig 2 clearer; Do the numbers on the x axis represent the total number of cases?

Why is the genotype composition so different between 2022 und 2023?

How have you calculated the divergence rates? Please mention that in the Methods section!

Please describe and explain the results detailed in the text!

What is the main message?

I don´t understand paragraph line 234 to 245. Clearly state in the results section, what portion of immunoassay positive samples you successfully sequenced. Have only 28% of samples NoV positive in the immunoassay confirmed positive by PCR? “Genetic Diversity and Phylogenetic Analysis for Human 2 Norovirus Infection Agents in Specific Municipalities within The Sverdlovsk Region of Russia”  is not correct

I would rather write:

“Genetic Diversity and Phylogenetic relationship of human norovirus sequences derived from Municipalities within The Sverdlovsk Region of Russia”

In line 90,91 you state: “Positive biological samples were confirmed through laboratory testing using enzyme immunoassays”

With immunoassays you detect only samples with higher viral loads (ca. > 10e5 Geq/ml). This means that your samples are selected

What percentage of the immunoassay positive samples have you successfully sequenced?

Please describe Fig 2 clearer; Do the numbers on the x axis represent the total number of cases?

Why is the genotype composition so different between 2022 und 2023?

How have you calculated the divergence rates? Please mention that in the Methods section!

Please describe and explain the results detailed in the text!

What is the main message?

I don´t understand paragraph line 234 to 245. Clearly state in the results section, what portion of immunoassay positive samples you successfully sequenced. Have only 28% of samples NoV positive in the immunoassay confirmed positive by PCR? “Genetic Diversity and Phylogenetic Analysis for Human 2 Norovirus Infection Agents in Specific Municipalities within The Sverdlovsk Region of Russia”  is not correct

I would rather write:

“Genetic Diversity and Phylogenetic relationship of human norovirus sequences derived from Municipalities within The Sverdlovsk Region of Russia”

In line 90,91 you state: “Positive biological samples were confirmed through laboratory testing using enzyme immunoassays”

With immunoassays you detect only samples with higher viral loads (ca. > 10e5 Geq/ml). This means that your samples are selected

What percentage of the immunoassay positive samples have you successfully sequenced?

Please describe Fig 2 clearer; Do the numbers on the x axis represent the total number of cases?

Why is the genotype composition so different between 2022 und 2023?

How have you calculated the divergence rates? Please mention that in the Methods section!

Please describe and explain the results detailed in the text!

What is the main message?

I don´t understand paragraph line 234 to 245. Clearly state in the results section, what portion of immunoassay positive samples you successfully sequenced. Have only 28% of samples NoV positive in the immunoassay confirmed positive by PCR?

Explain why 72% have been rotavirus positive! Do the screening tests detect both GI viruses?

Please refine the conclusion section! Which are the consequences of your data? Most therapeutic and preventive measures will not depend on phylogenetics of genotypes.

Small language correction:

Adult´s samples were not collected (line 266)

Comments on the Quality of English Language

Generally the English quality is fine, some sentences are difficult to understand, therefore I recommend moderate editing

Author Response

Comments from the Editors and Reviewers:

Reviewer 2: the authors analyzed norovirus sequences derived from stool samples of infants with AGE in specific municipalities of Russian Sverdlovsk region. Molecular NoV epidemiology data of that regions have not been published yet.

  1. Genetic Diversity and Phylogenetic Analysis for Human 2 Norovirus Infection Agents in Specific Municipalities within The Sverdlovsk Region of Russia is not correct

Response:

We agree with the correction and thus the title of the article has been modified to: “Genetic Diversity and Phylogenetic relationship of human norovirus sequences derived from Municipalities within The Sverdlovsk Region of Russia”.

  1. In line 90,91 you state: “Positive biological samples were confirmed through laboratory testing using enzyme immunoassays” With immunoassays you detect only samples with higher viral loads (ca. > 10e5 Geq/ml). This means that your samples are selected. What percentage of the immunoassay positive samples have you successfully sequenced?

Response:

We thank the reviewer for this question. As suggested by the reviewer, this has been changed in different sections of the manuscript in material and methods, results and discussion and highlighted in yellow in the article. The text now reads:

“All received biological samples were screened for the presence of HuNoV by using an enzyme immunoassay (Norovirus-antigen-enzyme immunoassays-Best, Vector-Best, Novosibirsk, Russian Federation) or qPCR (AmpliSens® Rotavirus/Norovirus/Astrovirus-FL, Central Research Institute of Epidemiology, Moscow, Russian Federation)”. (Materials and Methods Lines 91-94)

“A total of n=510 samples of clinical material were analyzed during the study period (n= 440 positive by enzyme immune-assay and n=70 positive by qPCR)”. (Results Lines 151-152)

“The percentage of successfully typed samples confirmed by ELISA was low (n=153/440, 34%), while the percentage of successfully typed samples confirmed by qPCR (n=43/70, 61%) was much higher”. (Discussion Lines 265-266).

  1. Please describe Fig 2 clearer; Do the numbers on the x axis represent the total number of cases?

Response:

We thank the reviewer for this point. Figure 2 describes the distribution of the HuNoV genotypes in the Sverdlovsk region in 2022-2023 per genogroup (circular charts) and per genotype (histogram charts) (a – The distribution of HuNoV in 2022, b – The distribution of HuNoV in 2023). The numbers on the x axis do not represent the total number of cases. The units along the x-axis show the number of typed samples for a specific genotype per municipality. The figures have been modified accordingly and the total number of samples were added on top of each figure (Lines 163-166).

  1. Why is the genotype composition so different between 2022 und 2023?

Response:

The significant variation in the genotypic profile during the analyzed period, spanning from 2022 to 2023, could stem from the post-COVID era and the anti-epidemic measures in place. It's noteworthy that the dominant genotypes GII.4 and GII.17 are consistently reported annually as they were reported before the COVID-19 pandemics. In the post-COVID era, the less dominant genotypes were more influenced. Genotypes categorized under GI might indicate cases imported from elsewhere. Continuing molecular-genetic surveillance in the Sverdlovsk region is essential for a more objective evaluation of the norovirus landscape and tracking trends in the circulation patterns of specific genotypes.

  1. How have you calculated the divergence rates? Please mention that in the Methods section! Please describe and explain the results detailed in the text!

Response:

We have calculated the divergence rates for GI.3, GII.4 and GII.17 strains by using the genetic analysis MEGA software, version 11. The divergence rates for GI.3/3039 is shown in the results section (Lines 181-184), and discussed later on in the text (Lines 279-287).

This text has been added to the material and methods section and highlighted in yellow The text now reads (Lines 140-146):

“The assessment of evolutionary divergence between sequences was conducted using the genetic analysis MEGA software, version 11. Distance estimation was performed on amino acid sequences aligned with the ClustalW algorithm. The model for determining genetic distance involves examining the proportion of nucleotide or amino acid sites (p/a) where two compared sequences are different. This comparison is achieved by dividing the number of p/a differences by the total number of compared p/a, expressing the divergence as a percentage”

This text has been added to the results section and highlighted in yellow (Lines 181-184):

“During amino acid sequence alignment, seven missense mutations were detected in genotype GI.3/3039 compared to selected reference sequences of GI.3 from Ekaterinburg. Analysis of the genetic distance matrix showed a significant 8.1% divergence in the amino acid sequences of GI.3/3039.”

  1. Please refine the conclusion section! Which are the consequences of your data? Most therapeutic and preventive measures will not depend on phylogenetics of genotypes.

Response:

The conclusion section has been refined and the following sentence has been added also highlighted in yellow (Lines 328-331).

“Routine monitoring is crucial for identifying circulating HuNoV genotypes that may subsequently be included in vaccine formulations”.

The data obtained from this study and other similar studies will allow to monitor and to establish a collection of strains that can be used in future studies to develop new treatments and vaccines.

  1. Have only 28% of samples NoV positive in the immunoassay confirmed positive by PCR? Explain why 72% have been rotavirus positive! Do the screening tests detect both GI viruses?

Response:

In our study, different ELISA kits were used to screen for HuNoV and rotavirus. We were concerned by the high proportion of untyped HuNoV ELISA-positive samples. Thus, 45 un-typed samples by ELISA were screened by real-time PCR using the AmpliSens® Rotavirus/Norovirus/Astrovirus-FL reagent kit (Central Research Institute of Epidemiology in Moscow, Russian Federation). HuNoV was detected in only 13 samples, while rotavirus was found in 32 samples. These findings suggest that the commercial immunoassay used may have had low specificity and lead to false-positive results for norovirus.

  1. I don´t understand paragraph line 234 to 245.

Response:

Since its first detection, the GII.4 genotype has been the dominant genotype worldwide. This explains why the monitoring of this genotype and other genotypes is important for the surveillance and the containment of Norovirus infection. New recombinant viruses or new genovariants with high contagiousness may appear in the coming future.